# The Comparative Effect of Lactic Acid Fermentation and Germination on the Levels of Neurotoxin, Anti-Nutrients, and Nutritional Attributes of Sweet Blue Pea (*Lathyrus sativus* L.)

**DOI:** 10.3390/foods12152851

**Published:** 2023-07-27

**Authors:** Nimra Arshad, Saeed Akhtar, Tariq Ismail, Wisha Saeed, Muhammad Qamar, Fatih Özogul, Elena Bartkiene, João Miguel Rocha

**Affiliations:** 1Department of Food Science and Technology, Faculty of Food Science and Nutrition, Bahauddin Zakariya University, Multan 60800, Pakistan; nimraarshad090@gmail.com (N.A.); tariqismail@bzu.edu.pk (T.I.); wishasaeed1@gmail.com (W.S.); muhammad.qamar44@gmail.com (M.Q.); 2Department of Seafood Processing Technology, Faculty of Fisheries, Cukurova University, 01330 Adana, Turkey; fozogul@cu.edu.tr; 3Biotechnology Research and Application Center, Cukurova University, 01330 Adana, Turkey; 4Department of Food Safety and Quality, Veterinary Academy, Lithuanian University of Health Sciences, Tilzes Str. 18, LT-47181 Kaunas, Lithuania; elena.bartkiene@lsmuni.lt; 5Institute of Animal Rearing Technologies, Faculty of Animal Sciences, Lithuanian University of Health Sciences, Tilzes Str. 18, LT-47181 Kaunas, Lithuania; 6CBQF—Centro de Biotecnologia e Química Fina—Laboratório Associado, Escola Superior de Biotecnologia, Universidade Católica Portuguesa, Rua Diogo Botelho 1327, 4169-005 Porto, Portugal; 7LEPABE—Laboratory for Process Engineering, Environment, Biotechnology and Energy, Faculty of Engineering, University of Porto, Rua Dr. Roberto Frias, s/n, 4200-465 Porto, Portugal; 8ALiCE—Associate Laboratory in Chemical Engineering, Faculty of Engineering, University of Porto, Rua Dr. Roberto Frias, s/n, 4200-465 Porto, Portugal

**Keywords:** grass pea, processing, germination, fermentation, nutritional composition, minerals, amino acids, phytic acid, tannin, β-ODAP

## Abstract

Grass pea (*Lathyrus sativus* L.), an indigenous legume of the subcontinental region, is a promising source of protein and other nutrients of health significance. Contrarily, a high amount of β-N-oxalyl-l-α,β-diaminopropionic acid (β-ODAP) and other anti-nutrients limits its wider acceptability as healthier substitute to protein of animal and plant origin. This study was aimed at investigating the effect of different processing techniques, viz. soaking, boiling, germination, and fermentation, to improve the nutrient-delivering potential of grass pea lentil and to mitigate its anti-nutrient and toxicant burden. The results presented the significant (*p* < 0.05) effect of germination on increasing the protein and fiber content of *L. sativus* from 22.6 to 30.7% and 15.1 to 19.4%, respectively. Likewise, germination reduced the total carbohydrate content of the grass pea from 59.1 to 46%. The highest rate of reduction in phytic acid (91%) and β-ODAP (37%) were observed in germinated grass pea powder, whereas fermentation anticipated an 89% reduction in tannin content. The lactic acid fermentation of grass pea increased the concentration of calcium, iron, and zinc from 4020 to 5100 mg/100 g, 3.97 to 4.35 mg/100 g, and 3.52 to 4.97 mg/100 g, respectively. The results suggest that fermentation and germination significantly (*p* < 0.05) improve the concentration of essential amino acids including threonine, leucine, histidine, tryptophan, and lysine in *L. sativus* powder. This study proposes lactic acid fermentation and germination as safer techniques to improve the nutrient-delivering potential of *L. sativus* and suggests processed powders of the legume as a cost-effective alternative to existing plant proteins.

## 1. Introduction

Worldwide, more than 50 edible legumes of varying shapes, sizes, textures, and colors have been recognized [1]. Globally, legumes and their products—being less expensive source of dietary vegetable proteins and minerals as compared to meat, egg, and fish—are considered potential sources of protein for humans [2]. There has been mounting evidence that excessive consumption of animal-based foods for the sake of proteins might enhance the risk for cardiovascular diseases in aged people [3,4]. Therefore, plant-based proteins could be an alternate strategy to diminish the health risks associated with animal-based foods, since plant-based protein sources also provides micronutrients and fiber besides being a plentiful source of proteins [5].

Grass pea (*Lathyrus sativus*) has been used as animal fodder and as a grain legume in human diet since the Neolithic period [6]. This indigenous crop’s seeds constitute 18–34% of protein content and 58% of PUFA [7,8]. This legume contains65% of carbohydrates and is thought to be a good source of minerals [9]. Grass pea (GP) represented 3.00 ± 0.10 g/100 g of ash, 25.6 ± 0.2 g/100 g of protein, 1.67 ± 0.18 g/100 g of lipids, and 72.91 ± 2.95 g/100 g of carbohydrate levels [10]. Vitamin A, B, and C and β-carotene has been reported in different *L. sativus* varieties [11]. The *L. sativus* grains are a rich source of essential amino acids (EAAs), such as lysine (lower in cereals), but contain a lower content of methionine and cysteine [12,13]. The deficiency of methionine, which plays a vital function in cardiovascular disease (CNS), can be tackled with a healthy diet comprising both cereals and legumes [14,15]. As the only dietary source of l-homoarginine that is beneficial in central nervous system (CVD) remediation [16,17], GP is considered a promising functional food in cardiac health [18].

Most legume grains contain biologically active compounds that may anticipate anti-nutritional properties and hinder nutrient assimilation [19]. The consumption of legumes is restricted mainly due to the co-existence of anti-nutrient factors (ANFs) [2]. Major ANFs present in GP include phytic acid, tannins, and β-N-oxalyl-l-α,β-diaminopropionic acid (β-ODAP) [20,21]. Phytate is mostly found in the stored form of phosphorus in several legumes and grains, reducing phosphorus assimilation and also forming insoluble networks with calcium, magnesium, iron, and zinc [22,23]. A higher intake of tannins is associated with compromised digestion and utilization of carbohydrates, minerals, and proteins by limiting the action of digestive enzymes [24]. Grass pea is also considered unsafe due to the presence of a lathyrogen, namely β-ODAP [25,26]. A higher consumption of GP is associated with a neurodegenerative disease called neurolathyrism [27]. Detoxifying the raw seeds of grass pea with simple processing operations such as soaking the grains in water, cooking the grains, and sprouting raw grains to decrease ANF content can somehow improve the consumer safety of GP [28]. 

A plethora of literature is available to demonstrate the role of different processing techniques to boost the nutritional significance of leguminous grains and to remediate the associated toxicant burden. A few of the most explored techniques include soaking, boiling, dehulling, autoclaving, fermentation, and germination [29,30]. Studies reported by Hotz and Gibson [31] validated that fermentation and malting could concurrently suppress ANFs and increase the availability of legume nutrients. Kaushik et al. [32] suggested germination as a cost-effective and commonly used process to improve the nutritional value of pulses, alongside enhancing the mean content of protein, vitamins, and dietary fibers, and improving mineral bioavailability.

A wave of awareness of the nutritional significance of a variety of legumes has arisen among consumers, leading to an increased market demand for legume-based food products. Despite the fact that an ample amount of protein is available in legumes, the presence of ANFs lowers the nutritional value of the leguminous crops, making them an inferior choice for nutrients of human health significance as a food source among consumers. This fact further suppresses the tendency of farmers to grow the crop at relatively larger scales. Despite them being high value crops, the cultivation of non-conventional legumes has always remained a low priority of farmers from developing countries. Considering the toxicological and anti-nutritional aspects of non-conventional legumes, different processing methodologies such as boiling, germination, soaking, and fermentation are considered cost-effective solutions to improving the nutritional value of neglected indigenous legume crops, and to enhance crop adoptability among the farming communities.

This research paper planned to identify the comparative effect of non-invasive techniques, viz. soaking, boiling, germination, and fermentation, on the nutritional profile of GP, the reduction of the intrinsic toxicant burden in the raw seeds of GP, and to evaluate the response of grass pea processing on the amino acid concentration.

## 2. Materials and Methods

### 2.1. Raw Materials and Chemicals

Grass pea, locally known as Desi Matri in Pakistan, was purchased from the local market of Khanpur, District Rahim Yar Khan, Punjab, Pakistan. Peas were manually separated from the pods and shade-dried in open air at 40 °C during April 2021. All the chemicals and reagents used in this study were procured from the local supplier of Sigma Chemical Co., Ltd. Standards of calcium, sodium, magnesium, potassium, iron, and zinc were purchased from BDH Chemicals (Shanghai, China).

### 2.2. Preparation of Raw and Processed Grass Pea (GP) Powder

Peas were manually separated from the pods and green grass peas were processed with different techniques: soaking, boiling, germination, and fermentation. Sorting and cleaning of samples was carried out manually. All the stones and foreign practical was removed from grass peas before further processing. Briefly, dried grass peas (100 g) were steeped in water (500 mL) for 9 h at ambient temperature according to the method of Khalil et al. [33], and boiling (100 g in 500 mL) was completed at 100 °C for 3 h followed by the method of Audu and Aremu [34] with little modifications in the drying method (cabinet drying at 50 °C). The fermentation of GP powder was carried out adopting the procedures of Yang et al. [35] using the *Lactobacillus plantarum* strain that was prepared in MRS broth at 30 °C for 48 h. The biomass was centrifuged at 1200× *g* for 20 m and suspended in 0.9% saline water solution to obtain a bacterial cell count of 10^7^ CFU per mL. The 100 g sample was merged in 0.2 mL of the above-mentioned inoculum suspension, and the fermentation was carried out at 30 °C for 2 days. Germination was performed using the guidelines of Budryn et al. [36] by soaking the grains in water for 3 h. The grains were spread on a tray and covered with a wet cloth so that the grains were subjected to appropriate humidity conditions for 48 h. The sprouts were washed with distilled water after every 5 h so that the grains remain moisturized. Only distilled water was sprayed on the grains during sprouting periods. Germination was carried out in the dark. After processing, peas were dried in cabinet dryer at 50 °C and converted into powder with approximately 150 µm mesh size using a hammer mill. Both raw and treated grass pea flours were set aside at ambient temperature in ziploc polyethylene bags for performing the analysis.

### 2.3. Nutritional Composition

The nutritional profile of raw and treated GP powder was evaluated according to the methods of AOAC [37]. Moisture (925.10), ash (923.03), protein (920.87), fat (920.85), and fiber (32-10) were determined using standard analytical methods. In detail, the moisture content was measured by placing 5 g of accurately measured sample in a hot air oven for 24 h. After 24 h, the sample was weighed and measured and a difference in weight was noted. The ash content of the sample was measured by using a muffle furnace at 550 °C. Protein content was measured using the kjeldhal method. Digestion (25 mL H_2_SO_4,_ Digestion tablet), distillation (10 mL of 40% NaOH, 2 drops methyl red, and 25 mL of 4% boric acid), and titration (0.1 N H_2_SO_4_) of the sample were carried out. The fat content was measured by using n-hexane through a soxhlet apparatus. Likewise, fiber content was measured through acid (H_2_SO_4_) and base (NaOH) washing for 30 min. After both alkali and acid washing, the remaining residue was transferred into a Petri dish and dried in a hot air oven at 110 °C. After drying, the sample was cooled and weighed, and is expressed as W₁. The dried residue was transferred into the crucible and ignited in muffle furnace and weighed again as W₂. The fiber content was measured using following formula: Fiber (%) = W1 − W2/initial sample weight × 100(1)

The carbohydrate content was calculated on a dry-weight basis, while energy values were computed as follows: Carbohydrates (%) = [100 − (ash + fat + protein + fiber)](2)
Energy (kcal/g) = 9.00 (%fat) + 4.00 (% carbohydrates + % protein)(3)

Minerals were determined in GP after open vessel wet digestion using an inductively coupled argon plasma spectrometer (ICAP, DigiBlock 3000 ICP-MS) for calcium, sodium, potassium, magnesium, iron, and zinc estimation.

Samples of raw and processed GP powder were pre-digested with nitric acid. Sinkovic et al.’s [38] procedure was followed with modifications to discover the mineral content. Half a gram of GP powder was digested in 5 mL concentrated HNO_3_ at 95 °C for ninety minutes. Further, 4 mL of hydrogen peroxide (30%) was added to each tube and processed at 95 °C for 20 min. Samples were cooled for 20 min and the volume of the tube was brought to 50 mL with distilled water and mixed well. Afterwards, samples were transferred to ICP tubes for analysis. The ICP-MS was calibrated using distilled water as a blank, standards of pure elements mixed to known concentrations, and two maize reference samples.

### 2.4. Determination of Vitamin A and C Content

Jadoon et al.’s [39] procedure was adopted to evaluate the vitamin A content. Five grams of GP sample was taken and vitamin A was extracted two times with methanol (5 mL). Samples were mixed with 50% 1 mL of KOH and 2 mL of ethanol for saponification in glass tubes. Thereafter, solutions were incubated at 45 °C for 2 h in a water bath with continuous shaking and purging of nitrogen gas. Then, water (1 mL) was added and the sample was extracted with ether (5 mL). The organic layer was recovered and dried under a stream of nitrogen at 37 °C. The residue obtained was again mixed in methanol (5 mL) via vortexing and diluted further with methanol solution (5%) containing triton X-100. Absorbance was monitored at 620 nm with UV-Visible spectrophotometer (UV-Vis 3000, ORI, Darmstadt, Germany). A standard solution of vitamin A was prepared by mixing 5 mg/L of β-carotene. The vitamin C content of the GP powder was determined in accordance with the method adopted by Desai [40]. GP powder was extracted with 50 mL 5% metaphosphoric acid and 40 mL 10% acetic acid solution in a 250 mL conical flask. Filtrate was collected after the filtration of the solution through Whatman filter paper and further used for vitamin C determination. Filtrate was mixed with few drops of bromine solution. Thereafter, thiourea solution (a few drops) was added to remove excess bromine solution. 2,4-Dinitrophenylhydrazine solution was added to both the filtrate and all standards for the onset of the coupling reaction. All standards and filtrate solution were kept 37 °C for 3 h for the completion of the reaction. Afterwards, the cooling of the solutions was carried out in an ice bath and sulfuric acid was mixed into it (5 mL). Absorbance was checked at 491 nm using UV-Visible spectrophotometer (UV-Vis 3000, ORI, Darmstadt Germany). A standard solution of ascorbic acid was prepared by mixing ascorbic acid (50 mg) in 100 mL of distilled water.

### 2.5. Determination of Anti-Nutritional Factors

The phytate content of GP powder was determined according to the method of Wolfgang and Lantzach P [41]. An amount of 0.06 g of GP powder was extracted with 10 mL HCl solution (0.2 N) in a test tube. Thereafter, the extract (1 mL) was heated with 2 mL ferric solution (0.2 g Fe (NH4)_2_(SO_4_)_2_ with 2 N HCl and volume was made to 1 mL) for 30 min in a water bath. Then, 2,2-bipyridine (4 mL) was put into 1 mL extract and the contents were vortexed. The absorbance of the sample and phytic acid standards were determined at 519 nm with UV-Visible spectrophotometer (UV-Vis 3000, ORI, Darmstadt, Germany). Tannins were assessed via Schanderi [42]. A total of 0.5 g of sample (GP powder) was heated with 75 mL of distilled water for thirty minutes and centrifugation was completed for twenty minutes at 2000 rpm. Afterwards, supernatant (1 mL) was mixed with 5 mL Folin-Denis reagent and 10 mL Na_2_CO_3_ solution. The final contents of the flask were made to 100 mL with distilled water, and 30 min stay time was given. The absorbance of tannic acid standards (0–100 µg) and samples was checked at 700 nm on UV-Visible spectrophotometer to determine the tannin content. β-ODAP content was determined using the method of Rao [43]. The compound was extracted from the sample using the method described by Abegaz et al. [44]. Eighty milligrams of the sample was vortexed with H_2_O (8 mL) and the extraction of β-ODAP was completed at 40–45 °C. Afterwards, centrifugation was completed at 4000 rpm and the obtained supernatant (0.1 mL) was transferred to test tubes for hydrolysis by using 0.2 mL of 3 M KOH for 30 min in a boiling water bath to convert ODAP to DAP. After the neutralization of the hydrolysate, the volume was made up to 1 mL; then, 2 mL OPA reagent was added to form a colored adduct. The absorbance of the adduct was checked at 476 nm using a spectrophotometer after giving a 30 min stay time. β-ODAP concentration was computed by using a DAP standard curve and treated with OPA to form a colored adduct, as in the case of the sample, and analyzed spectrophotometrically.

### 2.6. Determination of Amino Acid Content

The amino acid composition of the GP powder was determined with amino acid analyzer in accordance with the method proposed by Schuster [45]. The hydrolysis of the samples (3 mg) was completed with 6 N HCl (50 mL) at 110 °C for 24 h in a vacuum. To eliminate the unneeded hydrochloric acid (6 N), the resultant hydrolysates were dried in a rotator evaporator at 40 °C in a vacuum. The dried residues were added to citrate buffer (pH 2.2) and the hydrolysate was filtered to obtain a clear solution. An aliquot of the sample was injected into the HPLC column (Shim-pack ISC-07/S1504 Na), while amino acid analysis was completed with an amino acid analyzer (RF-10 AXL, Shimadzu Corporation, Tokyo, Japan) equipped with a florescence detector (FLD-6A). Two reaction solutions were used, namely sodium hypochlorite and o-pthalaldehyde.

### 2.7. Determination of In Vitro Protein Digestibility (IVPD%)

The grass pea powder sample (200 mg) was mixed in 35 mL of pepsin solution. A total of 0.035 M HCl solution with pH 2.0 was used to prepare the pepsin (1.5 mg/mL) solution. The sample and pepsin solution was incubated at 37 °C for 2 h. Afterwards, centrifugation of the sample was completed for 15 min at 12,000× *g* at 4 °C. The residue obtained was mixed with 0.035 M HCl and re-centrifuged. The residues obtained were dried overnight at 40 °C. The nitrogen content of the dried pellet was estimated using the micro-Kjeldhal method. A blank was run without any enzymes [46]. The results for protein digestibility were computed in accordance with following formula: Protein Digestibility% =100 − Protein contents in dried residue × 100/Total Protein contents(4)

### 2.8. Statistical Analysis

The study was replicated twice and the results were presented as mean ± standard deviation (SD). The data derived were statistically analyzed using a completely randomized design (CRD). Means were compared with a least significant difference at *p* < 0.05 as the level of significance. Statistix 8.1 version was used to perform the data analysis.

## 3. Results

### 3.1. Effects of Processing on the Nutritional Profile of Grass Pea

The results of the nutritional evaluation, viz. moisture, ash, protein, fiber, fat, and carbohydrate, were obtained for raw and processed GP powder and are presented in Table 1. The data present the worth of the considerable outcomes of the used techniques upon the nutritional composition of grass pea. The moisture content is a key factor in determining the shelf-life stability of legume flours. In this study, the moisture levels of GP powder differed significantly (*p* < 0.05); this may be attributed to different processing treatments and conditions. The highest moisture levels were recorded in raw GP flour, while the lowest moisture was observed in fermented GP flour samples. The germination of grass pea attributed the highest ash content, whereas comparatively lower ash content was observed in fermented GP powder. The results indicated GP as a source of protein, and a significant increase in protein levels of germinated GP powder was observed followed by fermented powder, while the lowest was found in raw GP powder. The fiber content of GP powder also varied significantly (*p* < 0.05) owing to the different processing treatments. An appreciable increase in the concentration of fiber was observed in the germinated GP powder samples as compared to the raw GP powder. The fat concentration in the raw GP powder was 0.59%, whereas in the treated groups, the fat content was found to be highest in the germinated GP powder, followed by the soaked, boiled, and fermented GP samples. The processing methods applied to GP depicted significant increases and decreases in the carbohydrate content of the product. Carbohydrate levels were found to be lowest in the germinated flour of the GP, followed by the fermented, soaked, and raw powder samples of GP as compared to the boiled samples. The caloric value of the raw GP powder was significantly (*p* < 0.05) higher, i.e., 331.8 Kcal/100 g), followed by the boiled, soaked, fermented, and germinated samples. 

### 3.2. Mineral Composition

In the present research, the GP powder developed from fermented legume samples contained comparatively higher Ca, Na, Mg, K, Fe, and Zn levels than the raw and germinated GP (Table 2). Contrarily, the concentration of minerals in the germinated GP samples presented a significant increase in K and Zn but a decline in Ca, Mg, Na, and Fe. The soaked GP samples showed an increase in potassium and magnesium content and a decrease in calcium, sodium, iron, and zinc, respectively. The boiled samples showed no increase in minerals, but showed a decrease in Ca, Mg, K, Na, Fe, and Zn. The increase may be due to the loss of anti-nutritional factors (ANFs) into the decanted soaking water, thus enhancing mineral bioavailability, and the decrease may be due to the loss of divalent ions into the decanted soaking water.

### 3.3. Vitamin Content of Grass Pea

The vitamin content of GP is presented in Table 3. The results suggest a significant effect of the treatments on the vitamin content of the GP powder. In detail, the vitamin A content was recorded as the highest in the raw samples, followed by the germinated, fermented, soaked, and boiled samples, respectively. Likewise, the vitamin C (ascorbic acid) content was found to be higher in the raw samples, followed by the germinated, soaked, boiled, and fermented samples, accordingly. It can be seen that, among the treatments, vitamin A and C content was found to be the highest in the germinated (T3) samples of GP, but lower than the raw GP samples. 

### 3.4. Anti-Nutrient Composition of Grass Pea

The anti-nutritional factor content of GP powder is elucidated in Table 4. The results suggested that all processing treatments anticipated a significant reduction in the phytic acid and tannin content of GP powder. In our study, the germination, fermentation, soaking, and boiling treatments deployed a 92, 86, 62, and 34% reduction in the phytic acid content of GP as compared to the raw GP powder samples. Moreover, the fermentation, germination, soaking, and boiling treatments deployed a 89, 87, 63, and 28% reduction in the tannin content of GP as compared to the raw GP powder samples. β-ODAP levels varied significantly among the treated GP powder samples with the different processing methods. Germination and fermentation deceased β-ODAP levels by 37 and 30%, respectively. In addition, a 21 and 4% reduction in β-ODAP due to soaking and boiling was also observed. The in vitro protein digestibility (IVPD) of raw, soaked, boiled, germinated, and fermented powdered samples are shown in Table 4. The percentage of IVPD of raw GP (74.95%) increased to 76.36% after soaking, 78.75% after boiling, 79.64% after germination, and 81.02% after fermentation.

### 3.5. Amino Acid Composition of Grass Pea

Leguminous crops are an affordable source of dietary protein and essential amino acids (EAAs) for vegetarian people with a sensitivity toward animal protein, and for socio-economically weaker communities. Different processing techniques like fermentation have the ability to increase the protein content and free amino acid concentration in legumes and legume-based products [47]. The results tabulated in Table 5 present the amino acid content of the GP powder samples. The results suggested that GP processing had a notable (*p* < 0.05) effect on the amino acid levels of the processed GP powder samples.

Our results indicated that the fermentation treatment was found to be effective in presenting higher concentrations of EAAs, i.e., isoleucine, leucine, threonine, histidine, lysine, and tryptophan. Among the listed essential amino acids, the highest amount of threonine, i.e., 8.67 g/100 g, was recorded in the germinated GP, while arginine levels (5.73 g/100 g) were found to be higher in the soaked samples. In the present study, methionine was found at 0.17 g/100 g in the raw GP sample, followed by a decline in the processed GP powder boiled sample (0.21 g/100 g). The lowest amounts were observed in the fermented GP powder sample, i.e., 0.08 g/100 g. The processing techniques also presented an improvement in the tyrosine, tryptophan, and phenylalanine content of the GP samples. They showed an increase in threonine (4.23 to 4.32 g/100 g), proline (0.37 to 0.42 g/100 g), glycine (0.65 to 0.76 g/100 g), leucine (1.85 to 1.96 g/100 g), isoleucine (0.66 to 0.85 g/100 g), aspartic acid (2.92 to 3.15 g/100 g), glutamic acid (1.80 to 1.82 g/100 g), tyrosine (1.07 to 1.26 g/100 g), tryptophan (0.03 to 0.14 g/100 g), lysine (0.51 to 1.03 g/100 g), and histidine (0.21 to 0.92 g/100 g) content.

## 4. Discussion

In the present study, the germination of grass pea produced the highest ash content, which is in line with the findings of Masood et al. [48]. On the other hand, the lower ash content recorded in the fermented grass pea powder could be related to mineral leaching or due to the uptake of minerals by the fermenting microbes for their proper growth [49]. Similar results were noticed in the soaked GP powder treatment [50]. The results corroborate the findings of Hanbury et al. [6] and Urga et al. [12], suggesting a 2.6–3.9 and 1.28–4.14% increase in the ash content of GP powder. The protein level in the raw GP was measured as 22.6% and was consistent with the research outcomes of Urga et al. [51]. A notable increase in the protein levels of the germinated GP powder was observed, while the lowest was found in the raw GP powder. Kumitch [52] reported that the fermentation process enhances the protein content in legumes and cereals. An increase in protein content as a result of germination has also been documented by Kavitha and Parimalavalli [53], where the protein content of germinated seeds increased from 21.9 (in raw) to 31.8% in mungbean flour. Similarly, the study reported that germination increased the protein content of germinated groundnut flour, i.e., from 29 to 32%. It was found by Bueno et al. [54] that the elevation in protein content as a result of germination could be attributed to hydrolysis and the release of embryonic proteins required for seed germination. This sort of change is not noticed in normal seeds. One of the principal functions of germination is to stimulate the activity of hydrolytic enzymes, which are found in an inactive state in ungerminated seeds [55]. These enzymes carry out metabolic activities through the breakdown of complex compounds, i.e., carbohydrates, proteins, and lipids, into simple and soluble molecules [56]. For example, proteases bring about the hydrolysis of proteins and are found in endosperm, the outer layer of endosperm (aleurone layer), and in germ, while amylase is found in pericarp and causes the breakdown of starch. The lipoxygenase enzyme is involved in the breakdown of lipids and is present in the embryo of seed, while other enzymes, namely peroxidase and polyphenol oxidase, which are present in bran portion, cause the hydrolysis of phenols. The increase in germination days leads to the enhanced nutritional value of leguminous and cereal grains, creating a positive effect on human health [57].

The germinated GP powder was recorded as having more fiber as compared to the raw GP powder, in line with the findings of Megat et al. [58] that showed higher levels of fiber, i.e., 72.4 and 59.9%, in germinated soybeans and kidney beans as compared to raw grains (32.01 and 36.5%). The GP fiber levels in this study ranged from 15.09 to 19.75%, which are much higher as compared to the previous findings of Urga et al. [12,51], who reported 3.6–8.6% fiber in grass pea. Such variations were probably due to the difference in varieties of GP. The release of new polycarbohydrates, i.e., starch and cellulose, as a result of germination [59], or the increase in the cellular structure of plants during germination may be the reason for the increase in fiber levels [60].

The fat concentration in the raw GP powder, i.e., 0.6–0.8%, is in line with the findings of Campbell [61]. Earlier, Hanbury et al. [6] reported higher levels of fat (0.9–5.3%) as compared to the present study. Among the treated samples, the fat content was found to be highest in the germinated GP powder, followed by the soaked and boiled samples, and the lowest was found in the fermented GP samples. This elevation of fat content may link to the release of fat as an energy source by new sprouts as a result of germination [62]. The reason for the slight increase in fat content in the boiled GP powder may be due to the breakdown of carbohydrate–lipid and protein–lipid linkages, leading to an increased oil recovery rate [50]. A decline in the level of fat in the fermented GP sample, as compared to the germinated sample, may be associated with the utilization of fat as an energy source for microbes (particularly lactic acid bacteria, LAB) for carrying out fermentation [63,64].

The levels of carbohydrates in the present study found in the raw GP powder samples, i.e., 59.1%, were on par with the boiled samples (60.67%). This is closely related to the findings of Urga et al. [12,51], who found 52.4 to 65.2% of carbohydrates in raw GP. Kavitha and Parimalavalli [53] reported a decline in the carbohydrate levels (61.2%) of germinated mungbean flour as compared to raw mungbean flour (65.1%), which supports the findings of the present study. The reason for this reduction may be due to the enzymatic activity of α-amylase that facilitates the synthesis of complex sugars into simple sugars which are used as source of energy in seed germination [65]. Olagunju et al. [66] found that the microorganisms (*Bacillus* spp.) that carry out the fermentation process are responsible for the production of different enzymes, such as glucosidase, lactanase, amylase, and fructofuranosidase, which break down the carbohydrates, leading to their reduction. The microorganisms that carry out fermentation use glucose as a preferred substrate, leading to a reduction in CHO content [67]. The reduction in CHO in the soaked GP powder (53.0%), compared to the raw powder (59.1%), might be attributed to the loss of soluble sugars in the soaking water [50]. In the present study, lower caloric values were observed in the germinated (T_3_) grass pea powder (309.068 Kcal/100 g), which may be due to the breakdown of starch during respiration that provides the energy to carry out metabolic activities in germinating seed [68]. The present study, in the context of caloric values, is also supported by Kavitha and Parimalavalli [53], who revealed higher energy values in raw groundnut flour (582.13 Kcal/100 g) as compared to germinated groundnut flour (574.16 Kcal/100 g).

Legumes are considered a promising source of minerals and, simultaneously, anti-nutritional factors. Mineral chelation with ANFs is the main hinderance to improve the bioaccessibility of minerals, especially divalent ions. Certain conventional processing techniques, such as steeping, soaking, fermentation, and germination, are found to be effective in increasing mineral bioaccessibility and bioavailability [31,69]. In the present research, fermentation led to an increase in Ca, Na, Mg, K, Fe, and Zn levels, in line with the findings of Olagunju et al. [66]. They reported a similar trend indicating higher mineral content such as Ca (0.43–0.48 mg/kg), Mg (0.39–0.43 mg/kg), Zn (88–95 mg/kg), P (0.97–1.06 mg/kg), Fe (33–44 mg/kg), Cu (20–24 mg/kg), and Mn (50–58 mg/kg) in fermented tamarind seeds, family leguminoseae. Another study by Onwurafor et al. [70] reported that fermentation caused varying responses to the concentration of minerals such as calcium (21.5–23.4 mg/100 g), iron (43.4–53.3 mg/100 g), and zinc (3.8–3.05 mg/100 g). This increase in mineral content can be ascribed to the breakdown of the anti-nutrient–mineral-chelation complex in the fermenting medium, due to the metabolic activity of fermenting microbes [70], and may also be linked to the reduction in dry-matter content owing to the breakdown of proteins and carbohydrates [71]. Conversely, in the present study, germination had varying effects on GP minerals in a way that the levels of K and Zn increased, while Ca and Mg decreased. Earlier, a study by Sangoris and Machado [72] reported that germination anticipated varying responses to the concentration of specific mineral elements in *Phaseolus vulgaris* and *Cajanus cajan*. A similar trend was observed by Laxmi et al. [59], suggesting that the germination of wheat (Fe; 4.9 to 4.65 mg/100 g; Ca: 48 to 26 mg/100 g; P: 355 to 212 mg/100 g), chickpea (Fe: 4.6 to 5.4 mg/100 g; Ca: 202 to 124 mg/100 g; P: 312 to 176 mg/100 g), and foxtail millet (Fe: 2.8 to 3.05 mg/100 g; Ca: 31 to 17 mg/100 g; P: 290 to 149 mg/100 g) modulated macro and micro element concentrations. Such a trend may likely be associated with varying soaking times and the freeing of bound minerals in complex matrices. Germination led to an elevation in macro and micro element concentrations and may also be linked to the decrease in anti-nutritional factors that bind minerals [73,74] due to activation of the endogenous phytase enzyme that breaks down phytic acid, thus making a larger amount of minerals bioaccessible [75,76]. Ungureanu-Iuga et al. [77] also documented an improved mineral profile of different legumes as a result of germination.

In our study, low levels of vitamin A were found in fermented samples (390 ± 8 µg/L) as compared to germinated samples (520 ± 9 µg/L), and a similar trend was reported by Onwurafor et al. [70], suggesting that fermentation causes a reduction in the vitamin A content of mungbean (265 to 105 µg/100 g). The results also correlate with the findings of Laxmi et al. [59], who reported an increase in vitamin C due to germination in wheat (0 to 6.2 mg/100 g), chickpea (3 to 7.1 mg/100 g), and foxtail millet (0 to 4.1 mg/100 g). The disruption of starch caused by the enzymes, i.e., amylases and diastases during germination, lead to an increase in glucose content that serves as a source for the development of ascorbic acid [78]. A reduction in vitamin content in the fermented (T_4_) grass pea sample can be ascribed to the loss of water-soluble vitamins in the liquid phase of the fermenting medium or broth [79], and it also depends on the strain of microbe used in fermentation and its metabolic actions. Vitamin C levels were found to be lowest in the boiled GP powder (169.5 ± 7 µg/L), followed by the soaked samples (215 ± 5 µg/L), as compared to the raw samples (246.6 ± 9 µg/L). This may likely be due to the water-soluble nature of this vitamin. Ascorbic acid (vitamin C) is a very unstable vitamin and can easily be denatured by heating and by soaking due its high solubility in water [80,81].

In the present study, the phytic acid in the raw GP powder (438.32 mg/100 g) was found to be lower than the results reported by Ramachandran [82], where 6520 mg/100 g phytic acid content was recorded in the extruded GP samples. Our results indicated that the germination treatment caused the highest reduction in phytic acid content (38.65 mg/100 g), when compared to the raw GP samples (438.32 mg/100 g). A reduction in phytate content was recorded at 74% in lentil sprouts as compared to the raw dry seeds of the lentil [83]. Earlier, research carried out by Van Vo et al. [84] reported that LAB fermentation decreased the phytate content in lupine seeds, which is consistent with the results of the present study. Soaking and boiling also resulted in the reduction of phytic acid content; a similar trend indicating lower phytate levels caused by soaking and boiling in pigeon pea was reported [85] due to phytate ions leaching into the soaking medium as an outcome of blanching, either steam or submerged blanching, and soaking. Lower levels of tannins were identified in the fermented GP powder (48.52 mg/100 g), while germinated, soaked, and boiled GP powder samples contained 59.36, 159.49, and 308.75 mg/100 g of tannins. Germination and fermentation reported 86 and 89% tannin content reduction. During LAB fermentation, the microorganisms stimulate the activity of the tannase enzyme that causes the breakdown of tannins [86]. Tannin losses occur due to its leaching into decanted fermenting media because of its water solubilization or evaporation during boiling [87]. Earlier, a finding by Coda et al. [88] indicated that Lactiplantibacillus plantarum induced fermentation to cause a reduction in tannin levels in faba bean flour. During soaking, the enhanced activity of the polyphenol oxidase enzyme causes the losses in tannins reported by Khandelwal et al. [89]. The tannin content was reduced from 5.67 to 4.07 mg/100 g due to germination in soy bean [90]. In the present study, β-ODAP levels vary significantly among the treated grass pea powder samples with different processing methods. The raw GP powder contained 427.17 mg/100 g β-ODAP, which is in line with the amount reported by Fikre et al. [91], i.e., 20 to 540 mg/100 g. The current investigation suggested a 37, 30, 21, and 4% reduction in β-ODAP due to germination, fermentation, soaking, and boiling, respectively, which is in agreement with the previous studies by Kumar et al. [92], who reported that soaking GP grains in water and boiling and roasting them at 150 °C greatly reduces the β-ODAP content. Indeed, Kuo et al. [93] suggested fermentation as a promising strategy to reduce β-ODAP levels to an extent of approximately 90%. The existing pool of information on the role of legume processing like fermentation and soaking in reducing lathyrogen concentration is possibly linked to the bacterial conversion of the toxic compounds to soluble non-toxic small fractions. Similarly, the isomerization and enzymatic-hydrolysis-lead conversion of β-ODAP to comparatively less and non-toxic α-ODAP is another possible mechanism of lathyrogen concentration. The studies suggest germination, fermentation, soaking, and boiling to reduce β-ODAP by 30–90% [47,93]. 

Protein digestibility refers to how well your body can use a particular source of dietary protein. High digestibility leads to a better nutritional value than low digestibility. A reduction in anti-nutrient factor levels increases protein digestibility, whereas higher levels of ANFs enable bonding with amino acids, thus hindering the proteolytic process [94]. It has been documented by Samtiya et al. [95] that phytic acid forms complexes with protein degradation enzymes, namely proteases, thus causing a decrease in the percentage of in vitro protein digestibility. The results of the in vitro protein digestibility of GP powder of the current research are consistent with the research of Linsberger-Martinet et al. [96], which showed higher percentages of IVPD in processed peas as compared to raw peas. Processing techniques significantly varied the IVPD percentage of GP powder. Comparatively, increased protein digestibility was noticed in boiled GP powder samples (78.75%) as compared to the raw material (74.95%), which is in line with the findings of Sanchez-Velazquez et al. [97], suggesting a 75.13–84.42% increase in the protein digestibility of cooked red kidney bean as compared to unprocessed red kidney bean. The elevation in IVPD % during germination may be attributed to the increased proteolysis carried out via intrinsic hydrolytic enzyme activity, i.e., proteases, thus increasing protein digestibility and amino acid bioavailability [98]. Our findings are in accordance with the work of Y. Di et al. [99], who documented that germination caused an increased response in IVPD from 43.6 to 47.9% in sesame seeds. It has been founded by Mir et al. [100] that enhanced digestibility is an indicator of good quality proteins because the onset of proteolysis leads to an increase in the release of amino acids. It has been determined by Rathore et al. [101] that fermentation can be assigned to break down larger protein molecules into smaller ones due to the action of proteolytic enzymes, leading to increased levels of free amino acids, hence enhancing protein solubility. A similar trend indicating increased protein digestibility due to the lactic acid fermentation in lupine seeds was noticed by Bartkiene et al. [102].

In the present study, amino acid profiling revealed a notable amount in the raw GP powder and the effects of different treatments, i.e., germination, fermentation, soaking, and boiling. Previously, Singh and Rao [16] and Llorent-Martínez et al. [18] reported grass pea as a functional food because it is the only dietary origin of _L_-homoarginine. Methionine, a sulfur-containing amino acid, is lacking in grass pea, but grass pea holds comparatively higher levels of lysine [13]. Heating or boiling significantly (*p* < 0.05) reduced the levels of aspartic acid, tyrosine, serine, alanine, glycine, and glutamic acid as compared to the raw GP samples, which is relevant to the research of Wu et al. [103], who noticed an elevation in proline, glycine, aspartic acid, arginine, alanine, and glutamic acid in quinoa seeds due to heating. Grass pea powder prepared with the fermentation treatment was found to have a comparatively higher concentration of essential and non-essential amino acids as compared to the raw sample. This may be due to the activity of fermenting microbes (viz., LAB) which starts the process of proteolysis leading to the release of free amino acids [104]. Lactic acid fermentation involves different metabolic activities carried out by lactic acid bacteria, chiefly proteolysis (the hydrolysis of proteins and peptides to free amino acids), lipolysis (the breakdown of lipids), and glycolysis (the breakdown of mono and disaccharides). With LAB being one of the extensively utilized lactic starters in fermented foods, it also increases the nutritional value and flavors of foods through the production of aroma components. Because of the inability of LAB to manufacture many amino acids, they encourage the hydrolysis of proteins and peptides to produce free amino acids along with the release of secondary products such as enzymes and bacteriocins, causing an increase in the shelf-life of fermented products [105]. Modulation in amino acid concentrations may likely be attributed to the kind of microorganisms to cause the onset of fermentation and the type of legume to be fermented. An almost identical finding was noticed by Cabuk et al. [106] and Verni et al. [107], indicating that lactic acid fermentation enhances the EAAs in pea protein concentrates and faba beans. A study reported by Espinosa-Páez et al. [108] suggested that the fermentation of oats with oyster mushroom escalates amino acid levels. The data presented in Table 5 indicate the reduction in arginine, phenylalanine, methionine, serine, alanine, and valine content of the fermented GP powder when compared with the raw GP samples. Such a trend may likely be attributed to the fermenting organism (lactic acid bacteria) utilizing these amino acids as energy sources to grow, thus reducing the amino acids further [52]. In our study, an increase in threonine, glutamic acid, proline, histidine, tryptophan, and arginine content was reported due to germination. Germination involves the breakdown of complex proteins through the stimulation of protease enzymes [56]. A similar trend indicating an increase in amino acids upon the germination of commercial legumes (beans, peas, and lentils) was reported by Jan et al. [62].

## 5. Conclusions

Among non-conventional legumes, *L. sativus* is considered as a widely cultivated pea species of South East Asian origin that plays a significant role in meeting the dietary protein and energy requirements of impoverished populations. The nutrient-delivering potential of this rare legume is, however, compromised with intrinsic toxicants and nutrient inhibitors. In line with previous findings, one role of different invasive and non-invasive processing techniques is in improving the nutritional composition, palatability, and safety of the leguminous foods. Our work proposes a comprehensive comparison of cost-effective and culturally acceptable processing techniques including soaking, boiling, germination, and LAB fermentation on the food and safety aspects of *Lathyrus sativus*. The results not only showed a reduction in phytic acid, tannins, and β-ODAP in fermentation from 438.32 to 64.78 mg/100 g, 424.43 to 48.52 mg/100 g, and 427.17 to 297.94 mg/100 g, respectively, and in germination from 438.32 to 38.65 mg/100 g, 424.43 to 59.36 mg/100 g, and 427.17 to 271.85 mg/100 g, respectively, in processed legume powder, but also yielded plausible improvements in the amounts of an array of essential nutrients. Fermentation improved mineral bioavailability, as the fermented samples contained comparatively higher Ca, Na, Mg, K, Fe, and Zn levels than raw GP due to the breakdown of the anti-nutrient–mineral-chelation complex in the fermenting medium. This study also proposes fermented and germinated *L. sativus* powder as a low energy, high protein and fiber composition that can help mitigate a wide range of nutritional and health disorders in the economically vulnerable populations of the South Asian subcontinent. Hence, it could be a better option to supplement preparing protein-enriched complementary snack foods and baked products for fermented and germinated grass pea powder so as to enhance the nutrient uptake of consumers.

## Figures and Tables

**Table 1 foods-12-02851-t001:** Proximate analysis of grass pea powder (g/100 g).

Treatments	Moisture	Ash	Protein	Fiber	Fat	Carbohydrates	Calories (Kcal/100 g)
T_0_	1.0 ± 0.03 ^a^	2.66 ± 0.02 ^e^	22.61 ± 0.02 ^d^	15.09 ± 0.02 ^e^	0.57 ± 0.02 ^a^	59.05 ± 0.02 ^b^	331.77 ± 0.1 ^a^
T_1_	1.52 ± 0.02 ^b^	2.77 ± 0.02 ^d^	27.96 ± 0.02 ^c^	16.07 ± 0.02 ^c^	0.18 ± 0.01 ^d^	53.02 ± 0.06 ^c^	325.54 ± 0.1 ^c^
T_2_	0.866 ± 0.02 ^c^	2.83 ± 0.02 ^c^	20.43 ± 0.02 ^e^	15.88 ± 0.02 ^d^	0.19 ± 0.01 ^c^	60.67 ± 0.03 ^a^	326.11 ± 0.2 ^b^
T_3_	0.773 ± 0.02 ^d^	3.613 ± 0.02 ^a^	30.71 ± 0.02 ^a^	19.42 ± 0.02 ^b^	0.24 ± 0.02 ^e^	46.017 ± 0.05 ^e^	309.068 ± 0.1 ^d^
T_4_	0.667 ± 0.02 ^e^	3.08 ± 0.01 ^b^	28.72 ± 0.03 ^b^	19.75 ± 0.03 ^a^	0.13 ± 0.01 ^b^	48.32 ± 0.07 ^d^	309.33 ± 0.2 ^d^

Values with different letters in the same column are significant at *p* < 0.05, mean ± SD. T_0_ = raw, T_1_ = soaked, T_2_ = boiled, T_3_ = germinated, T_4_ = fermented. Calculated on a dry-weight basis as 100—(Ash + Protein + Fiber + Fat).

**Table 2 foods-12-02851-t002:** Macro and micro mineral profiling of grass pea powder (mg/kg).

Treatment	Ca	Mg	K	Na	Fe	Zn
T_0_	4020 ± 0.02 ^b^	3910 ± 0.01 ^c^	3570 ± 0.02 ^d^	3020 ± 0.01 ^b^	3.97 ± 0.01 ^b^	3.52 ± 0.01 ^b^
T_1_	3980 ± 0.01 ^c^	4080 ± 0.01 ^b^	3960 ± 0.01 ^c^	2430 ± 0.02 ^d^	2.98 ± 0.01 ^d^	3.13 ± 0.06 ^c^
T_2_	3020 ± 0.02 ^e^	2940 ± 0.01 ^e^	2050 ± 0.02 ^e^	2170 ± 0.01 ^e^	2.89 ± 0.02 ^e^	2.69 ± 0.02 ^d^
T_3_	3550 ± 0.02 ^d^	3020 ± 0.02 ^d^	4030 ± 0.03 ^b^	2860 ± 0.02 ^c^	3.07 ± 0.01 ^c^	3.57 ± 0.01 ^b^
T_4_	5100 ± 0.01 ^a^	5080 ± 0.01 ^a^	4970 ± 0.01 ^a^	4970 ± 0.02 ^a^	4.35 ± 0.01 ^a^	4.97 ± 0.01 ^a^

Values with different letters in the same column are significant at *p* < 0.05, mean ± SD. T_0_ = raw, T_1_ = soaked, T_2_ = boiled, T_3_ = germinated, T_4_ = fermented.

**Table 3 foods-12-02851-t003:** Vitamin A and ascorbic acid levels of grass pea powder (µg/L).

Treatment	Vitamin A	Ascorbic Acid
T_0_	610 ± 7 ^a^	246.6 ± 9 ^a^
T_1_	290 ± 5 ^d^	215 ± 5 ^b^
T_2_	217 ± 4 ^e^	169.5 ± 7 ^c^
T_3_	520 ± 9 ^b^	235 ± 7 ^a^
T_4_	390 ± 8 ^e^	156.5 ± 8 ^e^

Values with different letters in the same column are significant at *p* < 0.05, mean ± SD. T_0_ = raw, T_1_ = soaked, T_2_ = boiled, T_3_ = germinated, T_4_ = fermented.

**Table 4 foods-12-02851-t004:** Anti-nutritional factors and IVPD% of grass pea powder (mg/100 g).

Treatment	Phytic Acid	Tannin	Β-ODAP	IVPD^ɬ^ (%)
T_0_	438.32 ± 2.33 ^a^	424.43 ± 2.07 ^a^	427.17 ± 4.51 ^a^	74.95 ± 0.01 ^e^
T_1_	168.33 ± 1.01 ^c^	159.49 ± 4.37 ^c^	337.92 ± 2.10 ^c^	76.36 ± 0.02 ^d^
T_2_	291.42 ± 6.61 ^b^	308.75 ± 5.87 ^b^	410.75 ± 1.30 ^b^	78.75 ± 0.01 ^c^
T_3_	38.65 ± 2.73 ^d^	59.36 ± 1.97 ^d^	271.85 ± 2.56 ^d^	79.64 ± 0.01 ^b^
T_4_	64.78 ± 0.43 ^e^	48.52 ± 0.38 ^e^	297.94 ± 2.53 ^e^	81.02 ± 0.01 ^a^

Values with different letters in the same column are significant at *p* < 0.05, mean ± SD. T_0_ = raw, T_1_ = soaked, T_2_ = boiled, T_3_ = germinated, T_4_ = fermented. IVPD^ɬ^, in vitro protein digestibility.

**Table 5 foods-12-02851-t005:** Amino acid composition of raw and processed grass pea powder (g/100 g).

T	AA	Th	Se	GA	Pr	Gl	Al	Me	Va	Is	Le	Ty	Hs	Tr	Pa	Ly	Ar
T_0_	2.92 ± 0.02 ^c^	4.23 ± 0.02 ^c^	3.17 ± 0.01 ^a^	1.80 ± 0.01 ^b^	0.37 ± 0.02 ^d^	0.65 ± 0.02 ^c^	1.13 ± 0.02 ^a^	0.17 ± 0.01 ^b^	0.53 ± 0.02 ^a^	0.66 ± 0.01 ^c^	1.85 ± 0.01 ^b^	1.07 ± 0.01 ^c^	0.21 ± 0.01 ^e^	0.03 ± 0.01 ^c^	1.33 ± 0.01 ^b^	0.51 ± 0.01 ^d^	4.07 ± 0.01 ^d^
T_1_	3.37 ± 0.01 ^a^	3.92 ± 0.01 ^e^	2.96 ± 0.01 ^b^	1.65 ± 0.02 ^d^	0.41 ± 0.01 ^b c^	0.93 ± 0.02 ^a^	0.85 ± 0.07 ^c^	0.13 ± 0.02 ^c^	0.46 ± 0.01 ^b^	0.82 ± 0.01 ^b^	1.76 ± 0.02 ^c^	1.28 ± 0.01 ^a^	0.85 ± 0.01 ^c^	0.08 ± 0.01 ^b^	1.38 ± 0.01 ^a^	0.97 ± 0.01 ^b^	5.73 ± 0.01 ^a^
T_2_	2.64 ± 0.03 ^d^	3.97 ± 0.01 ^d^	2.99 ± 0.02 ^b^	1.71 ± 0.01 ^c^	0.39 ± 0.02 ^c d^	0.57 ± 0.01 ^d^	1.04 ± 0.02 ^b^	0.21 ± 0.01 ^a^	0.5 ± 0.01 ^a^	0.61 ± 0.01 ^d^	1.62 ± 0.01 ^d^	0.87 ± 0.01 ^d^	1.36 ± 0.02 ^a^	0.05 ± 0.01 ^c^	1.07 ± 0.01 ^d^	0.62 ± 0.01 ^c^	4.35 ± 0.02 ^b^
T_3_	1.66 ± 0.02 ^e^	8.67 ± 0.01 ^a^	1.87 ± 0.01 ^c^	1.86 ± 0.01 ^a^	0.47 ± 0.01 ^a^	0.23 ± 0.05 ^e^	0.75 ± 0.01 ^d^	0.13 ± 0.02 ^c^	0.32 ± 0.02 ^c^	0.36 ± 0.01 ^e^	1.13 ± 0.01 ^e^	0.54 ± 0.01 ^e^	0.40 ± 0.01 ^d^	0.09 ± 0.01 ^b^	0.75 ± 0.01 ^e^	0.25 ± 0.03 ^e^	4.29 ± 0.01 ^c^
T_4_	3.15 ± 0.02 ^b^	4.32 ± 0.02 ^b^	1.83 ± 0.02 ^d^	1.82 ± 0.01 ^b^	0.42 ± 0.01 ^b^	0.76 ± 0.02 ^b^	1.03 ± 0.03 ^b^	0.08 ± 0.01 ^d^	0.51 ± 0.01 ^a^	0.85 ± 0.01 ^a^	1.96 ± 0.01 ^a^	1.26 ± 0.01 ^b^	0.92 ± 0.01 ^b^	0.14 ± 0.03 ^a^	1.24 ± 0.01 ^c^	1.03 ± 0.02 ^a^	1.07 ± 0.01 ^e^

Values with different letters in the same column are significant at *p* < 0.05, mean ± SD. T, treatments. AA, aspartic acid. Th, threonine. Se, serine. GA, glutamic acid. Pr, proline. Gl, glycine. Al, alanine. Me, methionine. Va, valine. Is, isoleucine. Le, leucine. Ty, tyrosine. Hs, histidine. Tr, tryptophan. Pa, phenylalanine. Ly, lysine. Ar, arginine. T_0_ = raw, T_1_ = soaked, T_2_ = boiled, T_3_ = germinated, T_4_ = fermented.

## Data Availability

All data related to the research are presented in the manuscript.

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
