# Peer review of "The Comparative Effect of Lactic Acid Fermentation and Germination on the Levels of Neurotoxin, Anti-Nutrients, and Nutritional Attributes of Sweet Blue Pea (Lathyrus sativus L.)"

_foods, 2023, doi:10.3390/foods12152851_

Round 1

Reviewer 1 Report

The present study was aimed for evaluation the impact of soaking, boiling, germination and fermentation on nutritional profile, amino acids composition and  intrinsic toxicants in raw seeds of grass pea.

methodology part is the weakest part of the manuscript. it would be difficult to repeat according the present description. For example:

- -line 165 - "GP powder was extracted with 50 ml metaphosphoric acid and acetic acid solution" - concentration of acids? Volume of acetic acid?

- -lines 202-203 - "Hydrolysis of samples was done with 6 N HCl" - amount of sample? Volume of the acid?

- -line 212 - "Grass pea powder sample (200 mg) was mixed in 35 ml of pepsin solution" - amount of pepsin?

- -all methodology must be reviewed and specified.

- use on term "sprouted" or "germinated" in the manuscript.

- it is not written how much water was used during soaking and boiling procedure, but is is important to evaluate which amount of nutritional compounds or amino acids passes to the water.

- explain, why the content of amino acids is higher then protein content?

- many of the data repeats both in text and in tables.

- add more details in the conclusions.

- Latin names must be in italic in the reference list.

Reviewer 2 Report

2.2. Preparation of raw and processed grass pea powder should be more detailed. If the provided references [33-35] have no DOI it is difficult to have access to these articles. Some more detailed regarding the methods applied for preparation of processed pea would be useful for readers. Eventually, the treatements hould be described as they are mentioned in the following tables: T1...T4. It would be interesting to have analyses for the fresh samples, before drying.

Nutritional composition (point 2.3.) should be detailed (briefly, mentioning only the method) in terms of the methods used for moisture, ash, protein, fat and fiber even if AOAC standard is given as reference.

In the Introduction, it is presented that pea grass contains lathyrogen. It is important to explain more, even if a reference is provided [27, 28] how the content of this toxic compound can be decreased and how much through soaking/germination/fermentation.

Round 2

Reviewer 1 Report

I agree with the improvements of the paper, just still minor corrections can be valuable:

- all formula must be numbered.

- I still see all data of vitamins A and C both in the text and table.
